# A prospective cohort study of cochlear implantation as a treatment for tinnitus in post-lingually deafened individuals
Qian Wang[1,2,3,4,7], Michelle R. Kapolowicz ●[5,6,7], Jia-Nan Li[2,3,4,7], Fei Ji[2,3,4], Wei-Dong Shen[2,3,4], Fang-Yuan Wang[2,3,4], Wei Chen[2,3,4], Wei-Wei Guo[2,3,4], Chi Zhang[2,3,4], Ri-Yuan Liu[2,3,4], Miao Zhang[2,3,4], Meng-Di Hong[1,3,4], Ai-Ting Chen[1,3,4], Fan-Gang Zeng ●[5] ✉ & Shi-Ming Yang[2,3,4] ✉

## Abstract

**Background** Cochlear implants have helped over one million individuals restore functional hearing globally, but their clinical utility in suppressing tinnitus has not been firmly established.

**Methods** In a decade-long study, we examined longitudinal effects of cochlear implants on tinnitus in 323 post-lingually deafened individuals including 211 with pre-existing tinnitus and 112 without tinnitus. The primary endpoints were tinnitus loudness and tinnitus handicap inventory. The secondary endpoints were speech recognition, anxiety and sleep quality.

**Results** Here we show that after 24 month implant usage, the tinnitus cohort experience 58% reduction in tinnitus loudness (on a 0–10 scale from 4.3 baseline to 1.8 = −2.5, 95% CI: −2.7 to −2.2, $p = 3 \times 10^{-6}$; effect size d' = −1.4,) and 44% in tinnitus handicap inventory (=−21.2, 95% CI: −24.5 to −17.9, $p = 1 \times 10^{-15}$; d'=−1.0). Conversely, only 3.6% of those without pre-existing tinnitus develop it post-implantation. Prior to implantation, the tinnitus cohort have lower speech recognition, higher anxiety and poorer sleep quality than the non-tinnitus cohort, measured by Mandarin monosyllabic words, Zung Self-rating Anxiety Scale and Pittsburgh Sleep Quality Index, respectively. Although the 24 month implant usage eliminate the group difference in speech and anxiety measures, the tinnitus cohort still face significant sleep difficulties likely due to the tinnitus coming back when the device was inactive at night.

**Conclusions** The present result shows that cochlear implantation can offer an alternative effective treatment for tinnitus. The present result also identifies a critical need in developing always-on and atraumatic devices for tinnitus patients, including potentially those with normal hearing.

## Plain language summary

Tinnitus is the perception that there is sound when it is not present. Cochlear implants are placed in the ears and can suppress tinnitus. However, the FDA do not yet recommend them as a tinnitus treatment. We evaluated 323 individuals with or without tinnitus before cochlear implantation and for over 2 years after implantation surgery. We investigated whether cochlear implantation is safe and effective for treating tinnitus and whether it causes tinnitus in people who did not have tinnitus previously. We found that cochlear implantation reduces tinnitus in 90% of those with pre-surgical tinnitus whilst causing tinnitus in only 3.4% of those without pre-surgical tinnitus. This finding confirms that cochlear implants are a safe and effective treatment for tinnitus.

As the most successful neural prosthesis, cochlear implants have restored functional hearing to one million individuals[1]. About half of the users are prelingually deafened children with most being able to develop normal language, while the other half are post-lingually deafened adults who can carry on a conversation on the phone[2,3]. Despite this success, treating hearing loss has remained the only approved indication for cochlear implantation since 1984.

Hearing loss is associated with many other ear disorders like tinnitus, which is the perception of sound that does not have an external source[4].

[1]Senior Department of Otolaryngology-Head and Neck Surgery, the First Medical Center of PLA General Hospital, Beijing, 100853, China. [2]Senior Department of Otolaryngology-Head and Neck Surgery, the Sixth Medical Center of PLA General Hospital, Beijing, 100048, China. [3]State Key Laboratory of Hearing and Balance Science, Beijing, 100853, China. [4]National Clinical Research Center for Otolaryngologic Diseases, Beijing, 100853, China. [5]Center for Hearing Research, Departments of Anatomy and Neurobiology, Biomedical Engineering, Cognitive Sciences, Otolaryngology—Head and Neck Surgery, University of California Irvine, Irvine, California, CA, 92697, USA. [6]Department of Communication Sciences & Disorders, University of South Florida, Tampa, FL, 33620, USA. [7]These authors contributed equally: Qian Wang, Michelle R. Kapolowicz and Jia-Nan Li. ✉e-mail: fzeng@uci.edu; yangshiming@301hospital.com.cn

Tinnitus is present in about 80% of those with hearing loss and 10-20% of the general population[5–7]. Tinnitus is a nuisance to some but debilitating to many, affecting focus and sleep, and leading to anxiety, depression and even suicide. Current standard care relies on behavioral therapies that relieve tinnitus symptoms, but do not cure tinnitus[8].

Historically, clinicians have noted that the cochlear implant could suppress tinnitus[9,10]. Further case studies identified electrical stimulation parameters for tinnitus suppression[11–13] and several clinical trials demonstrated promising results including treating incapacitating unilateral tinnitus in patients with single-sided deafness[14–18]. A 2021 review identified 10 studies with only 89 patients using intracochlear electrical stimulation to relieve tinnitus[19]. Despite the lack of a pre-defined protocol that resulted in >50% missing data, a large retrospective study of 300 patients also found a beneficial effect of the cochlear implant on alleviating tinnitus and distress[20]. A more recent prospective study involving 72 patients who responded to questionnaire showed that 58% of them complained about tinnitus before implantation and 60% of those with pre-implantation tinnitus experienced a clinically signification reduction 3 months post-implantation[21]. However, these studies have not provided sufficient evidence to make cochlear implantation a labeled indication for the treatment of tinnitus[22]. Additionally, reports on cochlear implantation inducing or worsening tinnitus have introduced outcome uncertainty and heightened the potential risk of implantation[23–25].

Here we designed a prospective two-year cohort study to examine the efficacy and risk of cochlear implantation on tinnitus and related symptoms in 323 post-lingually deafened individuals (>16 years old), including those with the pre-existing tinnitus ($n = 211$) and those having no tinnitus ($n = 112$). The main inclusion criterion for cochlear implantation was severe or more bilateral hearing loss (PTA > 80 dB HL) without chronic middle ear infections or significant lesions of the auditory nerve or brainstem. In the present study, individuals with congenital deafness were excluded because of the low prevalence of tinnitus, while children younger than 16 years old were excluded for their difficulty in reporting tinnitus[26,27]. We first established baseline measures before cochlear implant surgery, then obtained longitudinal measures from device activation to the 24 month post-activation endpoint. The primary endpoints were tinnitus loudness estimates on a 0–10 scale and tinnitus handicap inventory (THI)[28]. These primary endpoints determined not only the cochlear implant benefit for suppressing the pre-existing tinnitus but also its risk of inducing tinnitus in those without tinnitus. The secondary endpoints were speech recognition, anxiety and sleep quality, measured by Mandarin monosyllabic words[29], Zung Self-rating Anxiety Scale[30] and Pittsburgh Sleep Quality Index[31], respectively. The secondary endpoints determined whether and how pre-implant tinnitus and post-implant tinnitus changes affect cochlear-implant speech recognition and tinnitus-related symptoms like anxiety and insomnia[32–34].

We find that cochlear implants are safe and effective in tinnitus treatment, reducing tinnitus in 90% of those with pre-surgical tinnitus whilst causing tinnitus in only 3.4% of those without pre-surgical tinnitus. Two different time-scaled mechanisms are responsible for the observed tinnitus reduction: A fast (~minutes) mechanism modulated by the device turning-on or off, and a slow (~months) one controlled by the long-term device usage. We also find that not only does tinnitus not impact cochlear-implant speech performance, but more importantly the cochlear implant reduces tinnitus-related anxiety and insomnia.

## Methods
### Study design
We designed a prospective, mixed-longitudinal cohort study to examine the effect of cochlear implantation on tinnitus (Fig. 1). The mixed design

**Fig. 1 | Participant numbers and groups across study.** Cochlear implant candidates based on exclusion and inclusion criteria, tinnitus screening (Tinnitus or T-cohort vs. No-Tinnitus or NT-cohort), and intervention (unilateral or simultaneous bilateral cochlear implantation). *A major reason for "others" was invalid responses to the survey.

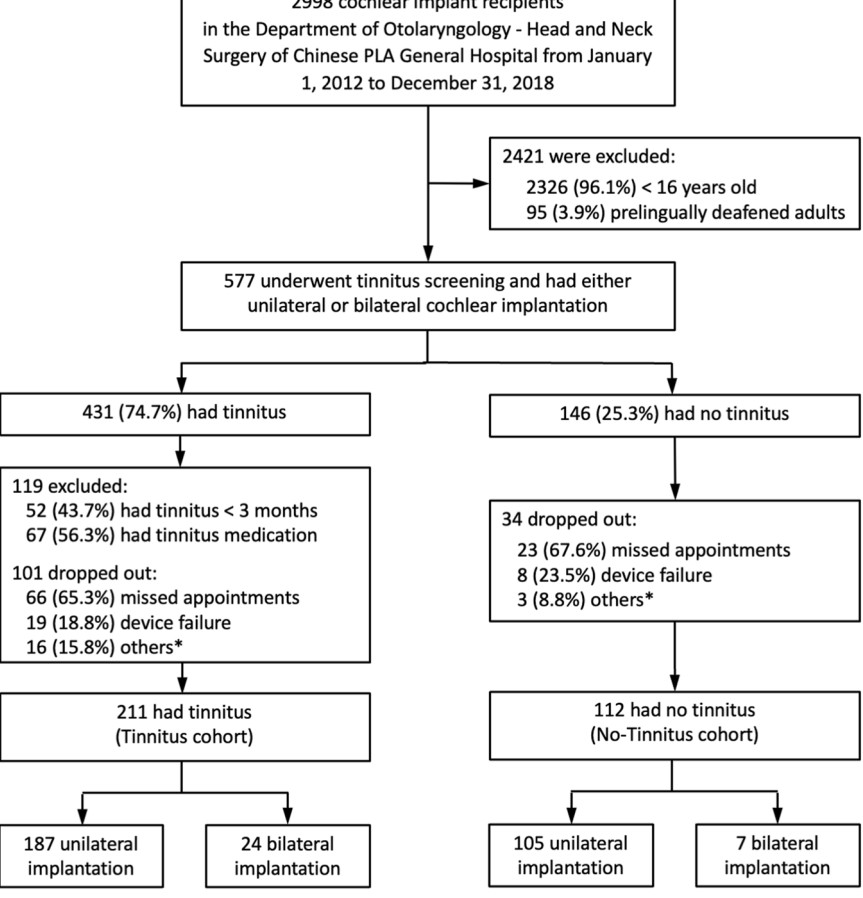

included an experimental group with chronic tinnitus (≥3 months) and a control group who reported no tinnitus before cochlear implantation. They all received the cochlear implants based on the usual standard of care. The physicians recommended cochlear implantation based on each patient's needs, while the patient decided whether to proceed with one or two cochlear implants and choose the type of the implant. Prelingually-deafened individuals were excluded due to their low prevalence of tinnitus[26]. Children (<16 years old) were also excluded due to difficulty in reporting tinnitus[27]. To minimize potential confounding factors, we excluded those who had acute tinnitus (<3 months) or received medication to treat tinnitus at the time of cochlear implantation. All eligible candidates with severe or more bilateral hearing loss (PTA>80 dB HL) in both groups underwent cochlear implantation surgery and the implant activation, which typically occurs about 4 weeks after the surgery when the wound is fully healed. The effects of cochlear implantation on tinnitus were measured before surgery to establish a baseline, on the day when the cochlear implant was activated and 1, 2, 3, 6, 12, and 24 months after activation (Fig. 2A). All participants gave written informed consent before taking part in the study.

During each visit from pre-surgery to 24 month post-activation, the participants were typically scheduled for a 2 h appointment at the clinic. The participants first underwent an activation or re-mapping session if needed. With the implant being turned on, they then performed the tinnitus loudness estimation and speech recognition tasks. After which, the implant was turned off for 30 min, and while the device was off, they completed the three questionaries (tinnitus, anxiety and sleep, see sections below) on paper under supervision at the clinic. They were instructed to rest or walk around if they finished the questionaries shorter than the 30 min. After this 30-min device off period, they performed the tinnitus loudness estimation task while the device was still off.

The present study was performed under a protocol approved by the Ethics Committee of Chinese PLA General Hospital (S2020-279-01) in accordance with the principles of Declaration of Helsinki and China FDA Good Clinical Practice. The study was registered with China National Clinical Trial Registry #ChiCTR2000035221.

### Primary endpoints: Tinnitus loudness and tinnitus handicap inventory

We selected two frequently used and easily administrated measures: tinnitus loudness estimate (range = 0–10 with 0 representing inaudible and 10 being an unbearably loud sound) and tinnitus handicap inventory (or THI, a 25-item questionnaire covering functional, emotional and catastrophic aspects of tinnitus; range = 0–100)[28]. Tinnitus loudness estimate takes seconds to complete by simply asking the subject "How loud is your tinnitus?", while THI takes 8–10 min to complete. Tinnitus loudness estimate is especially useful in monitoring time- and ear-sensitive changes in tinnitus. Here we measured dynamic changes in tinnitus loudness when the implant was not only turned on but also turned off for 30 min; we also measured ear-specific changes in tinnitus loudness due to a particular combination of the unilateral or bilateral cochlear implant stimulation vs. unilateral or bilateral tinnitus[12,35]. In contrast, the global THI measure cannot follow the short-term or ear-specific changes in tinnitus, but rather reflect the functional, emotional and catastrophic aspect of tinnitus disorder[36]. A reduction of 1.5 point on tinnitus loudness or 7 points on THI is considered clinically meaningful[37–39].

### Secondary endpoints: Speech, anxiety and sleep

Speech recognition was measured using Mandarin monosyllabic words[29] pre-surgically and on the implant activation day and follow-up visits. Also measured at the same time were the Zung Self-rating Anxiety Scale[30] and Pittsburgh Sleep Quality Index[31]. The anxiety rating is a norm-referenced test, with a score of 50 (range = 25–100) being considered as the threshold for anxiety disorder. The sleep index assesses sleep difficulty over a month-interval, with a score of 5 or higher (range = 0–21) being considered a "poor" sleeper.

### Statistical analysis

To compare group differences in baseline outcome measures, a Chi-squared test was performed when comparing the categorical variables of gender, etiology of deafness, and device type. For the remaining continuous variables, a Shapiro-Wilk test for normality was first performed. For normally distributed data (i.e., when the value of the Shapiro-Wilk test was >0.05), a two-sample two-tailed, t-test assuming unequal variance was used to determine group differences. For data that significantly deviated from a normal distribution, a Mann-Whitney U non-parametric test was used. A two-way, mixed ANOVA was used to the group difference in audiogram.

A one-way, repeated measures ANOVA was used to examine longitudinal effects of cochlear implantation usage on tinnitus and related symptoms. Adjusted p-values using the Dunnett's test were reported to correct for multiple post-hoc comparisons. An exponential function was used to fit all post-device activation data, with three free parameters being the baseline, asymptotic value, and time constant.

Cohen's d' was used to examine the effect size between the pre-surgery baseline and the 24 month post-activation result within the cohort or performance at the same time point between cohorts. The effect size is small for d' values within the range of 0.2–0.5, medium for 0.5–0.8 and large for ≥0.8.

Statistical analyses were performed using GraphPad Prism version 10.0 for Windows, GraphPad Software, Boston, Massachusetts, USA (www.graphpad.com).

## Results
### Characteristics of the participants
From January 1, 2012 to December 31, 2018, a total of 2998 patients received cochlear implants at the Chinese PLA General Hospital. 2141 were excluded based on either the age (younger than 16 years) or pre-lingual deafness criterion; the remaining 577 were then screened for tinnitus (Fig. 1). The screening showed that 431 of them (74.7%) had tinnitus, in which 119 individuals did not meet the inclusion criterion and 101 dropped out during the study. Thus, the tinnitus group (T-cohort) had a sample size of 211, with 187 of them receiving one cochlear implant while 24 receiving bilateral implants. The screening also showed that 146 (25.3%) had no tinnitus, in which 34 of them dropped out during the study. The no-tinnitus group (NT-cohort) had a sample size of 112, with 105 of them receiving one and 7 receiving two implants.

There were no significant group differences in gender, duration, etiology of deafness, cochlear implant brand and ears implanted (Tables 1, 2). There was also no significant group difference in the pre-surgical audiograms, showing similarly bilateral, symmetrical, sloping severe-to-profound hearing loss (Supplementary Fig. S1 and Supplementary Table S1). While the age range of cochlear implantation was identical (16 – 89 years old), the mean age was 4.5 years older in the T-cohort than the NT-cohort. There was also a significant group difference in baseline secondary endpoint measures, with the T-cohort having lower speech recognition (2.6% vs. 5.9% correct, p = 0.02), higher anxiety (50.2 vs. 43.2, $p = 2 \times 10^{-6}$) and greater sleep difficulty (7.6 vs. 4.8, $p = 1 \times 10^{-8}$). On average, the T-cohort had a clinically defined anxiety disorder (50.2 > threshold = 50) and insomnia (7.6 > threshold = 5).

Pre-operatively in the T-cohort, 139 (66%) had bilateral tinnitus and 72 (34%) had unilateral tinnitus, resulting in a total of 350 tinnitus ears, in which tinnitus loudness was independently measured (Table 3). The mean ± SD duration of tinnitus was 11.1 ± 10.7 years. Self-reported tinnitus types included constant multiple sounds (51%), constant tones (47%), and fluctuating sounds (2%). The severity of tinnitus before cochlear implantation was moderate regarding both tinnitus loudness (4.3 ± 1.6) and THI (48.7 ± 19.4).

### Primary efficacy endpoints
For the T-cohort, cochlear implant activation immediately reduced tinnitus loudness from the 4.3 baseline to 2.7 (difference: −1.6; 95% CI: −1.8 to −1.4; $p = 1 \times 10^{-15}$; effect size d' = −0.9), which was further reduced exponentially

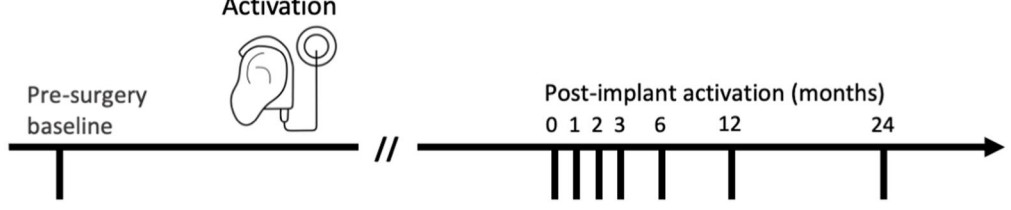

## A  Effects of cochlear implantation on tinnitus and related outcomes

Primary outcomes = Tinnitus loudness and Tinnitus handicap inventory (THI)
Second outcomes = Speech, Sleep and Anxiety

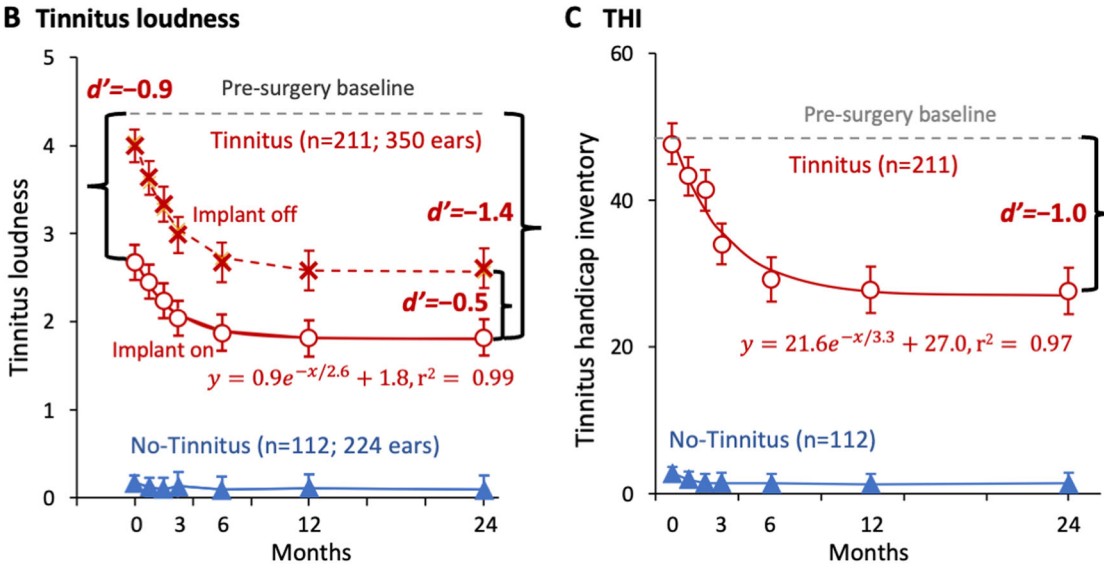

**B  Tinnitus loudness**

$d' = -0.9$

$d' = -1.4$

$d' = -0.5$

$y = 0.9e^{-x/2.6} + 1.8, r^2 = 0.99$

Tinnitus (n=211; 350 ears)

Implant off

Implant on

No-Tinnitus (n=112; 224 ears)

**C  THI**

$d' = -1.0$

Tinnitus (n=211)

$y = 21.6e^{-x/3.3} + 27.0, r^2 = 0.97$

No-Tinnitus (n=112)

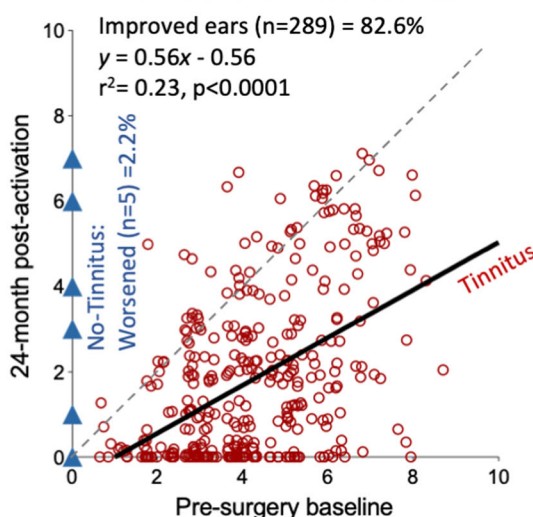

**D  Individual data for tinnitus loudness**

Improved ears (n=289) = 82.6%
$y = 0.56x - 0.56$
$r^2 = 0.23, p<0.0001$

No-Tinnitus: Worsened (n=5) =2.2%

Tinnitus

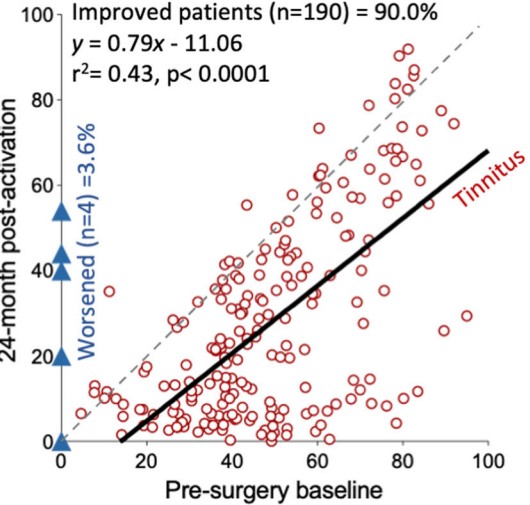

**E  Individual data for THI**

Improved patients (n=190) = 90.0%
$y = 0.79x - 11.06$
$r^2 = 0.43, p< 0.0001$

Worsened (n=4) =3.6%

Tinnitus

**Fig. 2 | Effects of cochlear implantation on tinnitus. A** In addition to the pre-surgery baseline, tinnitus and related symptoms were assessed at seven time points from device activation (0 month) to 24 months after. **B** Tinnitus loudness for the T-cohort (red circles = implant on; red-crosses=implant off for 30 min) and NT-cohort (blue triangles). Error bars represent the 95% confidence interval. The pre-surgical baseline was represented by a dashed horizontal line for the Tinnitus group and by the x-axis for the No-Tinnitus group. Also shown are the effect size (d') between the pre-surgical baseline and the implant on condition at 0 and 24 months, and that between the device on and off conditions. An exponential function was fitted to both the device on (red solid line) and off (red dashed line: $y = 1.4e^{-x/2.7} + 2.6, r^2 = 0.99$) conditions. **C** THI, symbols and lines were the same as in (**B**), except for a lack of the device off condition. **D** Individual tinnitus loudness data between pre-surgery baseline and 24 month post-activation endpoint for the T-cohort (circles) and NT-cohort (triangles, with the triangle on the origin representing data from 219 ears). Data points above or below the diagonal dashed line represent worsened or improved tinnitus loudness after cochlear implantation. The thick solid line represents linear regression for the Tinnitus group. **E** Individual THI data have the same representation as in (**D**).

**Table 1 | Demographic and hearing loss characteristics for the Tinnitus and No-Tinnitus cohorts**

| | Tinnitus (N = 211) | No-tinnitus (N=112) | Group differences | |
|---|---|---|---|---|
| **Gender** | **N(%)** | | | |
| Female | 103(49%) | 47(42%) | $X^2(1) = 1.0$ | $p = 0.32$ |
| Male | 108(51%) | 65(58%) | | |
| **Duration of deafness** | **Years** | | $t(230) = 1.7$ | $p = 0.14$ |
| Mean(SD) | 14.6(12.1) | 16.5(10.7) | | |
| Median | 12.0 | 15.0 | | |
| Range | 0.2-60.0 | 0.2-60.0 | | |
| **Etiology of deafness*** | **N(%)†** | | $X^2(8) = 7.9$ | $p = 0.44$ |
| Auditory neuropathy | 17(6%) | 2(2%) | | |
| LVAS & Mondini | 40(13%) | 23(20%) | | |
| Noise induced | 4(1%) | 2(2%) | | |
| Ossification & cochlear fibrosis | 6(2%) | 1(<1%) | | |
| Ototoxicity | 22(7%) | 9(8%) | | |
| Presbycusis | 22(7%) | 8(7%) | | |
| Sensorineural deafness | 127(42%) | 51(45%) | | |
| Sudden deafness | 53(17%) | 12(10%) | | |
| Other | 16(5%) | 7(6%) | | |

Categorical data were analyzed from the percentage values using Chi-squared tests. *Some reported >1 cause of deafness.

**Table 2 | Cochlear implant and baseline characteristics for the Tinnitus and No-Tinnitus cohorts**

| | Tinnitus (N = 211) | No-tinnitus (N = 112) | Group differences | |
|---|---|---|---|---|
| **Cochlear implant brand** | **N(%)** | | $X^2(3) = 5.7$ | $p = 0.13$ |
| AB | 47(13%) | 29(26%) | | |
| Cochlear | 122(35%) | 37(33%) | | |
| Med El | 124(36%) | 33(29%) | | |
| Nurotron | 57(16%) | 13(12%) | | |
| **Ears implanted** | **N(%)** | | $X^2(1) = 1.6$ | $p = 0.20$ |
| Unilateral cochlear implants | 187(89%) | 105(94%) | | |
| Bilateral cochlear implants | 24(11%) | 7(6%) | | |
| **Age of cochlear implantation** | **Years** | | $U = 9694$ | $p = 0.01$ |
| Mean(SD) | 40.3(15.9) | 35.8(16.0) | | |
| Median | 39.3 | 30.9 | | |
| Range | 16.2-89.4 | 16.0-89.3 | | |
| **Speech recognition** | **0–100% correct** | | $U = 1304$ | $p = 0.02$ |
| Mean(SD) | 2.6(3.5) | 5.9(8.7) | | |
| Median | 0.0 | 2.0 | | |
| Range | 0.0-14.0 | 0.0-40.0 | | |
| **Self-rating anxiety scale** | **25–100 scale** | | $U = 8031$ | $p = 2 \times 10^{-6}$ |
| Mean(SD) | 50.2(12.9) | 43.2(10.5) | | |
| Median | 49.0 | 43.0 | | |
| Range | 25.0-91.0 | 25.0-79.0 | | |
| **Pittsburgh sleep quality index** | **0–21 scale** | | | |
| Mean(SD) | 7.6(4.5) | 4.8(2.4) | | $U = 7269$ | $p = 1 \times 10^{-8}$ |
| Median | 7.0 | 4.0 | | |
| Range | 0.0-21.0 | 0.0-11.0 | | |

Mann-Whitney tests were used for non-normal data per Shapiro-Wilk results. Chi-squared tests were used for categorical data from percentages. Significant group differences are indicated by p values in bold text.

to an asymptotic level of 1.8 at the 24 month endpoint (total reduction: $-2.5$, 95% CI: $-2.7$ to $-2.2$; $p = 3 \times 10^{-6}$; d' = $-1.4$; $\tau$ = 2.6 months; red circles in Fig. 2B; statistical results in Supplementary Table S2). This total reduction of 2.5 points was equivalent to 58% decrease from the baseline tinnitus loudness. Interestingly, tinnitus came back after turning off the implant for 30 min (red crosses), and this significant "rebound" in tinnitus loudness parallelled the exponential decay of the device on loudness function ($y = 1.4e^{-x/2.7} + 2.6$; mean on vs. off difference = $-1.0$, $F_{1,349} = 284.6$, $p = 1 \times 10^{-15}$, d' = $-0.5$). Similar to the tinnitus loudness with the device off condition, THI did not show any immediate reduction at the implant activation but had a similar exponential reduction pattern (total reduction: $-21.2$, 95% CI: $-24.5$ to $-17.9$; $p = 1 \times 10^{-15}$; d' = $-1.0$; $\tau$ = 3.3 months; red circles in Fig. 2C; statistical results in Table S3). This total reduction of 21.2 points was equivalent to 44% decrease from the baseline THI. In contrast to the T-cohort, the NT-cohort had a constant minimum of 0.1 for loudness and 1.7 for THI (blue triangles in Fig. 2B, C; statistical results in Table S2 and S3), which was due to four NT-patients developing tinnitus after implant activation.

The overall risk of developing tinnitus after cochlear implant activation was low for the NT-cohort: Only five ears in four patients, i.e., 2.2% based on total ears or 3.6% on total patients (blue triangles above zero in Fig. 2D, E). All four patients had unremarkable demographic and audiological characteristics. Specifically, the four patients consisted of two females and two males, who were aged between 16 and 23 years, had typical etiologies (two with sensorineural deafness, one Mondini and one unknown), and received one cochlear implant. Three of them developed tinnitus on the implanted side 1 month after device activation, while the fourth one developed bilateral tinnitus immediately after activation.

For the T-cohort, cochlear implant activation and usage reduced tinnitus loudness in 82.6% of the 350 tinnitus ears (red circles below the diagonal line in Fig. 2D) and THI in 90.0% of the 211 patients (Fig. 2E). Linear regression (thick black line) showed not only a constant reduction (negative intercept in the equation) but also a benefit proportional to the baseline severity (<1 slope) for both loudness and THI measures. Specifically, the baseline tinnitus loudness was reduced by 30% or more in 260 out of 350 ears (74.3%) at 24 month post-implantation activation, including, importantly, total suppression or no tinnitus in 127 (36.3%) cases. Furthermore, 29 ears (8.3%) had tinnitus loudness reduction between 0 and 30%. In contrast, 63 ears (17.4%) had increased tinnitus loudness

**Table 3 | Tinnitus characteristics for the Tinnitus cohort**

| Tinnitus laterality | N (%) |
|---|---|
| Bilateral tinnitus | 139 (66%) |
| Unilateral tinnitus | 72 (34%) |
| **Tinnitus duration** | **Years** |
| Mean (SD) | 11.1 (10.7) |
| Median | 9.0 |
| Range | 0.25-52.0 |
| **Tinnitus type** (some reported >1 type) | **Occurrence (%)** |
| Monotone | 169 (47%) |
| Multiple sounds | 181 (51%) |
| Fluctuating | 8 (2%) |
| **Tinnitus loudness** (from 350 ears) | **0-10 scale** |
| Mean (SD) | 4.3 (1.6) |
| Median | 4.0 |
| Range | 1.0-9.0 |
| **Tinnitus handicap inventory or THI** | **0-100 scale** |
| Mean (SD) | 48.7 (19.4) |
| Median | 46.0 |
| Range | 2.0-98.0 |

All tinnitus characteristics were based on self-reports by the participants who filled a tinnitus survey form on paper.

post-implantation, including 55 (15.7%) with 0-30% increase and only 6 (1.7%) with >30% increase in tinnitus loudness. At 24 month post-implantation activation, the baseline THI was reduced by 31% or more in 130 out of 211 participants (61.6%) and by 0–30% in 60 participants (28.4%); the THI was increased in only 21 participants (10.0%), including 19 (9.0%) with 0–30% increase and 2 (0.9%) with >30% increase.

Analysis of the interaction between the device and tinnitus laterality showed that the cochlear implant suppressed tinnitus more effectively when both were on the same side than the opposite side (Fig. 3A, B; statistical results in Supplementary Table S4). For the 121 bilateral tinnitus participants who received a single cochlear implant, within-subjects comparison showed greater tinnitus suppression on the same than opposite side (mean difference: $-1.7$; 95% CI: $-2.3$ to $-1.1$; $p = 1 \times 10^{-15}$; d' = $-1.1$). The same pattern of result was also obtained in between-subjects comparison of 34 patients with the implant and tinnitus on the same side against 32 patients with the implant and tinnitus on opposite sides (mean difference: $-0.7$; 95% CI: $-1.9$ to 0.4; $p = 0.01$; $d' = -0.6$). Because of this same-side dominance, an additional contralateral implant did not produce any significantly more tinnitus suppression than the unilateral implant (Fig. 3C, Supplementary Table S4; mean difference: 0.1; 95% CI: $-0.5$ to 0.8; $p = 0.67$; $d' = 0.1$).

**Secondary efficacy endpoints**

Both the T- and NT-cohorts showed exponential improvement in speech recognition over time ($F_{6,477} = 267.20$, $p = 1 \times 10^{-15}$; Fig. 4A). Despite the slightly poorer baseline performance in the T- than the NT-cohort (2.6% vs. 5.9% correct, Table 2), there was no group difference with cochlear implant usage ($F_{1,477} = 0.53$; $p = 0.47$). After 24 month device usage, the T-cohort reached asymptotic performance of 79.0% (95% CI: 75.7 to 82.3%; $p = 1 \times 10^{-13}$; d' = 9.7; $\tau = 2.7$ months), while the NT-cohort reached 76.4% (95% CI: 70.4 to 82.4%; $p = 1 \times 10^{-15}$; d' = 6.0; $\tau = 2.7$ months).

The anxiety rating result had the same pattern as the speech result. Both showed exponential reduction in anxiety over time ($F_{6,1926} = 198.10$, $p = 1 \times 10^{-15}$; Fig. 4B). Despite the significant difference in baseline between the T- and NT cohorts (50.2 vs. 43.2, Table 2), there was no group difference in post-implantation measures ($F_{1,321} = 2.89$; $p = 0.09$). After 24 month device usage, the T-cohort reached an asymptotic anxiety rating of 38.8 (95% CI: 37.5 – 40.1; p = $1 \times 10^{-15}$; d' = $-1.0$; $\tau = 1.2$ months), while the

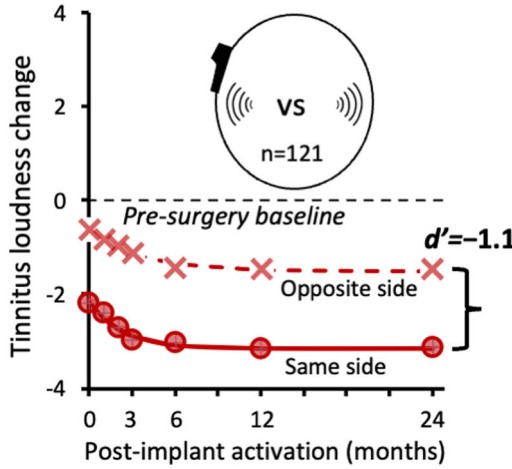

**A Effect of implant on bilateral tinnitus (within-subjects comparison)**

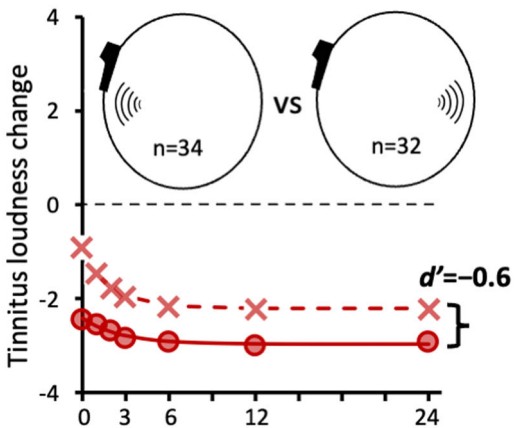

**B Effect of implant on unilateral tinnitus (between-subjects comparison)**

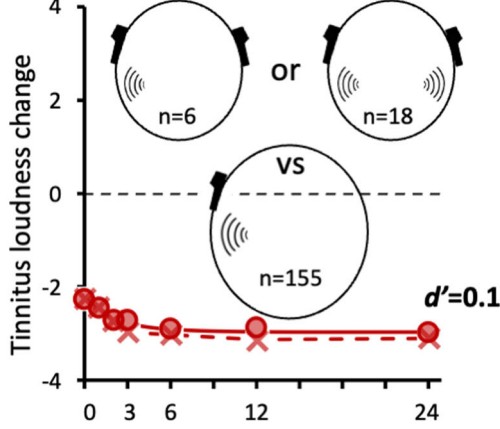

**C Effect of bilateral implants (between-subjects comparison)**

NT-cohort reached 36.6 (95% CI: 35.1 – 38.0; $p = 3 \times 10^{-14}$; d' = $-0.7$; $\tau = 2.3$ months). Notably, the long-term device usage reduced the T-cohort's anxiety below the clinical disorder threshold (=50; dashed horizontal line in Fig. 4B).

Like the speech and anxiety results, sleep quality was significantly improved with cochlear implant usage for both cohorts ($F_{6,1926} = 73.45$,

**Fig. 3 | Effects of cochlear implantation on ipsilateral and contralateral tinnitus.**
**A** Comparison of changes in tinnitus loudness (relative to pre-surgery baseline) caused by unilateral cochlear implantation in the same 121 bilateral tinnitus participants (crosses represent the cases where the implant and tinnitus were on the opposite side, while circles represent where the implant and tinnitus were on the same side). **B** The same representation as (**A**), except that the comparison was made in 34 ipsilateral tinnitus participants whose implant was on the same side as their tinnitus and 32 whose implant was on the opposite side to their tinnitus.
**C** Comparison between 24 participants who received simultaneously bilateral implants (crosses) and 155 who received a unilateral implant (from 121 bilateral tinnitus and 34 unilateral tinnitus participants), as shown in Panel (**A**, **B**).

$p = 1 \times 10^{-15}$; Fig. 4C). After 24-month device usage, the T-cohort reached an asymptotic sleep difficulty rating of 6.6 (95% CI: 6.1 to 7.2; $p = 1 \times 10^{-15}$; d'=−0.2; $\tau = 2.7$ months), while the NT-cohort reached 4.2 (95% CI: 3.7 − 4.6; $p = 2 \times 10^{-9}$; d' = −0.3; $\tau = 1.3$ months). In contrast to the speech and anxiety results, a significant group difference remained throughout the entire 24 month period ($F_{6,321} = 37.89$, $p = 2 \times 10^{-9}$; d' = 0.8). Despite significant reduction in tinnitus and related symptoms, the T-cohort was still classified as "poor sleepers" (=5; dashed horizontal line in Fig. 4C).

## Discussion

Over a 10 year span in a single site, we screened 577 post-lingually deafened individuals (>16 years) for cochlear implantation and found that 431 of them, a 75.3% prevalence rate, had pre-existing tinnitus (T-cohort) and 146 or 24.7% had no tinnitus (NT-cohort). We followed 323 patients for 2 years after cochlear implant activation (221 in the T-cohort; 112 in the NT-cohort). We observed an immediate reduction in tinnitus loudness at the device activation timepoint, followed by a large effect (d' = −1.4 and −1.0) of total reduction in both tinnitus loudness and handicap inventory at the 24 month endpoint. At an individual level for the T-cohort, 90.0% reported decreased THI scores, including 61.6% reporting >30% reduction and 28.4% reporting 0–30%, while only 10.0% reported increased THI post-implantation. Importantly, we found that the risk of cochlear implantation inducing tinnitus was low, as only four patients in the NT-cohort, or 3.6%, developed tinnitus. While the distributions varied, the present results, especially the relatively high percentage of total suppression and the low incidence of tinnitus induction cases, were consistent with recent studies[20,21,40].

The present result showed that cochlear implants can provide large improvements in tinnitus symptoms, which are comparable or greater than other state-of-the-art treatments. For example, hearing aids, or sound therapy, or their combinations reduced tinnitus functional index by 21-33 points after 4 months[41]. Tinnitus retraining therapy reduced tinnitus loudness by −1.8 and THI by −6.1 over 18 months[38], while cognitive behavioral therapy reduced the THI by −12.8 over 12 months[42]. Emerging treatments such as paired tone and vagus nerve stimulation reduced tinnitus loudness by −0.6 and THI by −20.2 over 3 months[43], while bimodal sound and tongue stimulation reduced THI by −13.5−14.6 over a 12 week treatment period[44], which was further reduced to −21.2 by changing parameter settings in the second 6 week period[45]. In contrast, drug therapy has so far produced either marginal or non-significant improvement[46]. Finally, we noted that the present improvement of −2.4 in tinnitus loudness and −21.2 in THI is remarkably similar to the −4.5 and −23.2 corresponding values from a meta-analysis of 27 studies reporting on 1285 patients[22].

We also found that long-term cochlear implant usage significantly reduced tinnitus-related symptoms while improving speech recognition. The quite large effect (d' = 6.0 and 9.7) in speech recognition was a testimony to the success of cochlear implantation for treating deafness as the main indication[3,47]. The medium-to-large effect size in anxiety reduction (d' = −0.7 and −1.0) in both cohorts was likely due to improved communication ability, whereas the disappearance of the initial baseline difference in anxiety was likely due to the sense of control over tinnitus in the

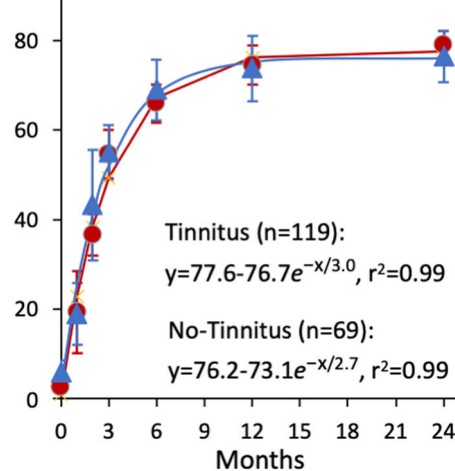

**A Speech recognition (%-correct)**

Tinnitus (n=119):
$y=77.6-76.7e^{-x/3.0}$, $r^2=0.99$

No-Tinnitus (n=69):
$y=76.2-73.1e^{-x/2.7}$, $r^2=0.99$

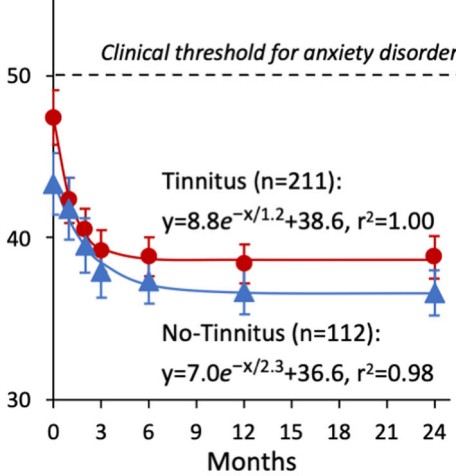

**B Anxiety rating (25-100 scale)**

*Clinical threshold for anxiety disorder*

Tinnitus (n=211):
$y=8.8e^{-x/1.2}+38.6$, $r^2=1.00$

No-Tinnitus (n=112):
$y=7.0e^{-x/2.3}+36.6$, $r^2=0.98$

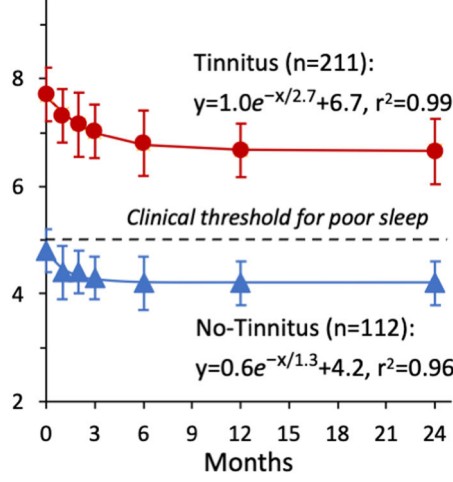

**C Sleep difficulty (0-21 scale)**

Tinnitus (n=211):
$y=1.0e^{-x/2.7}+6.7$, $r^2=0.99$

*Clinical threshold for poor sleep*

No-Tinnitus (n=112):
$y=0.6e^{-x/1.3}+4.2$, $r^2=0.96$

**Fig. 4 | Longitudinal measures of speech recognition and tinnitus related symptoms. A** Speech recognition for the Tinnitus (red circles) and No-Tinnitus (blue triangles) cohort. **B** Anxiety rating. The dashed line represents the clinical threshold for anxiety disorder (=50). **C** Sleep difficulty. The dashed line represents the clinical threshold for poor sleeper (=5). Error bars represent the 95% confidence interval.

T-cohort[48]. The small effect size (d'= −0.2 and −0.3) in reducing sleep difficulty was likely due to different reasons for the two cohorts. The NT-cohort did not have much sleep difficulty at the baseline, the small improvement was due to the floor effect. On the contrary, the T-cohort still had significant sleep difficulty even at the 24 month endpoint, reflecting that tinnitus can come back at the same level or higher when not wearing the cochlear implant (see red crosses in Fig. 2B).

While the present study provided strong support for using cochlear implants to treat tinnitus, it did not address the underlying mechanisms. Nevertheless, there is room for more benefit because current devices are optimized solely for speech recognition. There is also a critical need to develop novel and customized electric stimulation that can maximize tinnitus suppression while preserving speech performance on an individual basis[49–51]. Importantly, the present result identified an unmet need to produce an always-on cochlear implant that is small and consumes low power for patients to suppress tinnitus at night to improve sleep quality[48]. A hurdle for wide application of cochlear implantation to tinnitus suppression is that most tinnitus patients have residual or even normal hearing, which would likely be lost with current cochlear implantation devices and techniques. A solution for this group of tinnitus patients is to provide stepwise intervention from sound therapy to non-invasive electric stimulation, after which cochlear implantation using an atraumatic electrode array and a soft surgical approach to preserve hearing as a last resort[35]. If future implantation causes no or minimal hearing loss, then the cochlear implant can be an effective treatment for not only deafness but also tinnitus, even for those tinnitus sufferers who have significant residual or normal hearing.

## Data availability
The source data for Figs. 2, 3 and 4 can be found in Supplementary Data 1. Pure-tone audiometry data and statistical analysis results are provided in Supplementary Materials, including audiograms (Fig. S1) as well as statistical analysis of audiograms (Table S1), tinnitus loudness (Table S2), tinnitus handicap inventory (Table S3) and effects of one or two implants on unilateral or bilateral tinnitus (Table S4). All other data are available from the corresponding authors on reasonable request.

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

## Acknowledgements

We thank Harrison Lin, Yasmeen Hamza, Gerard O'Donoghue, and several anonymous reviewers for helpful comments on the manuscript, Dongyi Han, Pu Dai, and Hui Zhao for performing the cochlear implantation surgery, and the staff of the Auditory Implantation Center and Tinnitus Research Group of the Department of Otolaryngology for patient care. The work is supported in part by National Key Research and Development Program# 2022YFC2402702 (J.N.L.), National Key Research and Development Program# 2021YFF0702302 (J.N.L.), Youth Independent Innovation Science Fund Project #22QNFC032 (Q.W.), UC Irvine Center for Hearing Research (F.G.Z.) and National Institutes of Health 5R01 DC015587 (F.G.Z.). The funders of the study had no role in study design, data collection, data analysis, data interpretation, or writing of the article.

## Author contributions

Conceptualization: J.N.L., F.G.Z., S.M.Y. Methodology: Q.W., M.R.K., J.N.L., F.J., W.D.S., W.W.G., C.Z., R.Y.L., F.G.Z., S.M.Y. Investigation: Q.W., J.N.L., F.J., W.D.S., F.Y.W., W.C., M.Z., M.D.H., A.T.C., F.G.Z., S.M.Y. Data analysis: Q.W., M.R.K., F.G.Z. Visualization: Q.W., M.R.K., F.G.Z. Funding acquisition: J.N.L., G.W., F.G.Z. Project administration: Q.W., J.N.L., S.M.Y. Supervision: S.M.Y., F.G.Z. Writing—original draft: Q.W., F.G.Z., S.M.Y. Writing—review and editing: Q.W., M.R.K., F.G.Z. All authors had full access to the data and approved the final version of the article.

## Competing interests

The authors declare the following competing interests: F.G.Z. is a shareholder of Axonics, DiaNavi, NeoCortix, Nurotron, Syntiant, Velox and Xense. Other authors declare no competing interests.
