## [Transparent Peer Review file · Communications Medicine]

A prospective cohort study of cochlear implantation as a treatment for tinnitus in post-lingually deafened individuals

Corresponding Author: Professor Fan-Gang Zeng

Version 0:

Reviewer comments:

Reviewer #1

(Remarks to the Author)

1. Brief summary of the manuscript

The manuscript describes a prospective cohort study assessing the long-term effect of cochlear implantation on tinnitus distress in 323 adult patients with post-lingual deafness. Tinnitus loudness and THI were the primary outcomes. Speech perception, anxiety and sleep quality were the secondary outcomes. These outcomes were measured pre-implantation and at 1, 2, 3, 6, 12, and 24 months after activation. The study showed a significant reduction in tinnitus loudness and distress in the tinnitus cohort. Four patients experienced tinnitus induction after implantation. Speech recognition, anxiety, and sleep difficulties scores significantly decreased at 24 months post-activation compared to baseline. The tinnitus group on average still experience sleep difficulties.

2. Overall impression of the work

The manuscript showed the overall benefit of cochlear implant on tinnitus loudness, distress and related outcomes. The results are not novel (PMCID: PMC9515522, PMCID: PMC10605020), but the study adds value to the field and confirms the findings of the current literature on the effect of cochlear implant on tinnitus. Although the results are clear, the research questions need to be better defined and consistent with the data reported in the manuscript. The manuscript can be improved by providing more details on methods and reporting results (missing data).

3. Specific comments, with recommendations for addressing each comment

Abstract:

1. Line 27: Please specify here the questionnaire / measurement used to assess tinnitus loudness.
2. Line 31: Please specify the questionnaire / measurement used to assess speech, anxiety and sleep.

Introduction

1. Line 44: Please define what is tinnitus and add a reference to the definition.
2. Line 50: Suggest using the word "electrical" instead of "electric".
3. Line 50-57: Only case studies are mentioned here, whereas several works have been published on the topic. Suggest updating this section by referencing to recent systematic reviews on the effect of CI for tinnitus and systematic review on the effect of intracochlear electrical stimulation on tinnitus (e.g. PMC8615734).
4. Line 58: Safety is part of the main objective but there is no measurement of safety or reporting of harm and adverse events in the rest of the manuscript. Please remove the word "safety" from the objective or add safety data in the manuscript. Be consistent throughout the manuscript.
5. Line 65: Same as Line 27. Please specify here the questionnaire / measurement used to assess tinnitus loudness. VAS? Tinnitus matching test?
6. Line 67-68: Same as Line 31. Please specify the questionnaire / measurement used to assess speech, anxiety and sleep.
7. Line 69-70: The secondary aim is unclear. What is the objective? Please rephrase this sentence "The secondary measures would test whether tinnitus affects cochlear implant speech recognition and tinnitus reduction, if any, would alleviate tinnitus-related symptoms."

Results

1. Line 74: Was the age of 16 years old chosen as the threshold and not 18 (to select adult)?
2. Line 84: It could be interested to add information on the cohort hearing profiles (bilateral, unilateral, asymmetrical hearing loss; pure tone average).
3. Line 91-94: Specify that this is pre-operative data.
4. Line 105: Does it mean that patients without tinnitus were asked to fill the VAS and THI as well at each follow-up time point? I would suggest that instead of reporting the average of the NT-cohort, you only report the average score of the patient

with induced tinnitus.

5. Line 109-116: Was tinnitus induced right after activation for these 4 patients? Or was tinnitus induced at a later follow-up time point?
6. Line 117: Please be more precise in reporting results. Instead of "reduced" precise if this reduction is statistically significant and add the difference in score between baseline and 24 months post-implantation.
7. Line 132: "Despite the significant difference in baseline between the T- and NT cohorts (50.2 vs. 43.2, Table 1), there was no group difference ($F_{1,321}=2.89$; $P=0.09$)." Please specify that it is post-implantation.
8. General comments in the results section:
 - a. Were there patients reporting total tinnitus suppression?
 - b. According to Figure 2, there were cases of tinnitus worsening within the T-cohort. Please report the number of patients reporting worsening.
 - c. Suggest calculating how many patients reported a total suppression, clinically significant improvement, no significant change and clinically significant worsening. It will give a nice overview of the tinnitus outcomes in the cohort.

Discussion

1. Line 174: Suggest to better explain the situation CI on / CI off. Please don't use the word "increase" but explain that tinnitus can come back at the same level or higher when not wearing the CI.
2. Line 179: Suggest referencing the systematic review on the effect of intracochlear electrical stimulation on tinnitus (e.g. PMC8615734).
3. General comments in the discussion section:
 - a. Please write a paragraph on the observed changes in tinnitus after cochlear implantation in the study (worsening, induction, reduction) and compare it to the current literature (example of recent literature: PMID: PMC9515522, PMID: PMC10605020). Also, interesting to elaborate on why these changes are observed.
 - b. Please write a paragraph on the limitations of the study (missing data, VAS loudness not used in CI on and CI off, ...).
 - c. Several studies and systematic reviews have reported the same observations (e.g. PMID: 26153087), the authors should mention this in their conclusion.

Methods

1. Line 190-201: As not everyone is familiar with the hearing loss criteria for CI candidacy in China, I would suggest specifying it. Please specify if it is for bilateral and unilateral hearing loss and the degree of hearing loss for instance. Also interesting to specify why some candidates get one or two CIs, based on which criteria?
2. Line 190-201: Were there sequential bilateral implantations in the cohort? Did you take this into account?
3. Line 190: Remove 's' in the word "effects".
4. Line 194: Same as Line 74.
5. Line 201: Please specify the questionnaire administration type. Was it an online questionnaire or a paper questionnaire? Was it completed by the patient during his/her visit at the clinic or before?
6. Line 206: If the tinnitus loudness estimate is a Visual Analogue Scale (VAS), please mention it and keep this wording throughout the manuscript for consistency.
7. Was the VAS loudness used with the CI was on or off or in both conditions? Please specify.
8. Line 215: Suggest using the standard Minimal Clinically Important Difference defined in the literature as specified in the publications: 7 points for THI (PMID: 21493265) and 1.5 for VAS loudness (PMID: 22846637).
9. Line 223: Please specify with which software/tool you perform the statistical analysis.
10. Line 224-228: Did you test the normal distribution of your data?
11. Line 224-228: How did you manage missing data? Please report missing data at every follow-up time point.

Reviewer #2

(Remarks to the Author)

The authors provide a study that tracks and assesses quite a large number of cochlear implant patients over time in terms of their tinnitus, as well as other outcomes of speech perception, sleep and anxiety. The importance and high impact of the study is in the large numbers possible. Overall, the results are encouraging and important in showing that cochlear implant stimulation over time can improve tinnitus symptoms, as well as other outcomes that are expected based on previous studies. The paper is generally written in a clear way, however some key details are missing. Other suggestions to improve the manuscript are provided below.

>>There are multiple places where sentences and wording can be improved; please review carefully to catch all of them. Some examples are listed below:

-Abstract: Though anecdotal evidence suggests that cochlear implants suppress tinnitus, this utility is not yet established. In a decade-long study, we examined longitudinal effects of cochlear implants on tinnitus in 323 post-lingually deafened adults including 211 with pre-existing tinnitus and 112 without [tinnitus].

-Abstract: Prior to implantation, the tinnitus cohort had lower speech recognition, higher anxiety and poorer sleep quality than the non-tinnitus cohort, [in which] the 24-month implant usage eliminated this group difference in speech and anxiety measures.

-Introduction: Last 3 sentences say "would" but already happened so like previous sentences use past tense.

-Results, update to:

From January 1, 2012 to December 31, 2018, a total of 2998 patients received cochlear implants at the Chinese PLA General Hospital. 2141 were excluded based on either the age (younger than 16 years) or pre-lingual deafness criterion; the remaining 577 were then screened for tinnitus (Fig. 1). The screening showed that 431 of them (74.7%) had tinnitus, in which 119 individuals did not meet the inclusion criterion and 101 dropped out during the study. Thus, the tinnitus group (T-cohort) had a sample size of 211, with 187 of them receiving one cochlear implant while 24 receiving bilateral implants. The screening also showed that 146 (25.3%) had no tinnitus, in which 34 of them dropped out during the study. The no-tinnitus group (NT-cohort) had a sample size of 112, with 105 of them receiving one and 7 receiving two cochlear implants.

-Results, update to:

There were no significant group differences in gender, duration, etiology of deafness, cochlear implant brand and ears implanted (Table 1). There were also no significant group differences in the pre-surgical audiograms, showing similarly symmetrical, sloping severe-to-profound hearing loss (Fig. S1).

-Results section; formatting of CI can be improved, such as:

(total reduction= -21.2; 95% CI of, -24.5 to 0-17.9; $P < 0.0001$...) where need some spacings and filler words; also confusing to include % for mean value but not CI so remove or provide completely.

-Discussion, 3rd paragraph: Using the word "super-large effect size" is a bit odd; consider using "quite large effect size" or something more formal.

-Discussion last sentence could be improved to:

A solution for this group of tinnitus patients is to provide stepwise intervention from sound therapy to non-invasive electric stimulation, after which cochlear implantation using an atraumatic electrode array and a soft surgical approach to preserve hearing could be possible as a last resort. The present findings pave the way for cochlear implantation to be an alternative and effective treatment for not only deafness but also tinnitus.

-Figure 1 title could be more reflective of content such as, "participant numbers and groups across study", since not really showing study design (actual design and timeline is shown in Figure 1A).

>>One major improvement needed is in more rigorously describing the selection of the statistical analyses performed, such as why Chi-squared tests and in justifying normality assumption for the t-tests. Also, it wasn't sufficiently described how multiple comparisons issue was addressed, such as which endpoints were pre-specified or in what order, or that the multiple comparisons were addressed. Please provide further details to clarify and justify statistical approach. For example, claiming a significant group difference in baseline secondary outcome measures of T-cohort having lower speech recognition than NT-cohort with a $p = 0.02$ is not convincing when so many comparisons are being done.

>>Another major improvement needed is in better describing the study design and procedures in the Methods section, such as how were patients identified, what was done at each visit and what order of assessments, and even how were tinnitus characteristics determined (e.g., unilateral vs. bilateral, tinnitus type, etc.). Were they just asked questions or written on paper or a tablet, etc. Please provide more details in the first part of the Methods section for the reader to follow in detail what was done during the study. Also include the full inclusion/exclusion list in the Methods.

>>Please provide in the Methods the ethics approval information and number and that study was performed in accordance with those requirements.

>>The results are quite positive and convincing that cochlear implant improves tinnitus over time, and the results are well presented in the figures and greatly appreciate that the authors provided the individual participant data for tinnitus and loudness in Figure 2, which is convincing.

>>Note that MCID for THI is 7 points and not 6-7 points. Please update in Methods.

>>Multiple statements in Discussion need to be revised since they are not fully valid or clear.

-In paragraph 2, authors make strong claims that cochlear implant was "greater and faster" than other approaches. A major concern is the authors did not accurately or sufficiently cite data from previous studies. The vagus nerve study reported 17.7 for THI in terms of percentages and not points so those values need to be updated if comparing in points. The bimodal sound and tongue stimulation is based on treatment data for 12 weeks and not 6 months, in which there is a more recent study showing ~20 points by 12 weeks (actual values can be obtained from Conlon, B., et al. Different bimodal neuromodulation settings reduce tinnitus symptoms in a large randomized trial. *Sci Rep* 12, 10845 (2022)). So those data need to be updated. Also, there are several sound therapy studies showing comparable or even better results at similar rates (see papers by James Henry). Even if they don't use THI and they use TFI or other outcome measures, it doesn't mean they can be ignored for comparison (e.g., see Henry, J. A., et al., Tinnitus management: Randomized controlled trial comparing extended-wear hearing aids, conventional hearing aids, and combination instruments. *Journal of the American Academy of Audiology*, 28(06), 546-561). Based on the available literature, it appears cochlear implant can provide large improvements in tinnitus symptoms that can be better or comparable to other leading treatments that is as fast or in some cases faster for other types. The comparisons are more nuanced than the broad strong claim the authors make; please more carefully compare and explain the outcomes relative to many more studies to be valid.

-Last paragraph the authors claim "little research has been done" on cochlear implant effects on tinnitus improvements. This is not fully true in that multiple studies on this topic especially in unilateral deafness/tinnitus patients. Please adjust wording

and also cite some of these previous studies to put authors results into more comprehensive context.

-The authors should further think about the intention of the cochlear implant approach in their statement: "The present findings pave the way for cochlear implantation to be a safe and effective treatment for not only deafness but also tinnitus." Are the authors claiming individuals get implanted with a cochlear implant solely for tinnitus, even in individuals with some hearing remaining in that ear side?" That seems then there are major risks that can be possible such as losing their residual hearing and needs to be further discussed pros/cons. If only for those with sufficient hearing loss, it would be helpful to put that conclusion in the context the types of patients assessed in this study and how much hearing loss they had to identify generalizability of results to those with less hearing loss.

Reviewer #3

(Remarks to the Author)

This is a clinical trial on the effect of cochlear implants on a 211 hearing loss and tinnitus cohort. I have some concerns with the study design since the inclusion criteria are not clearly defined and there is no control group.

The experimental group included patients with > 3months with chronic tinnitus; however, there is no information about the psychoacoustic (tinnitus frequency, loudness, MML, residual inhibition), limited information about psychometric (THI score range 2-98) and most important hearing levels for low and high frequencies. It is expected that many of these variables can explain the outcome of the trial.

The inclusion criteria should be precisely defined: tinnitus type, defined THI range and defined threshold hearing. The data are presented as any tinnitus type and the authors should differentiate any tinnitus from tinnitus disorder (see reference), since they are clinically different in term of management and interventions.

According to the information provided, no control group (tinnitus patients without CI) or randomization method was used. Please, try to clarify this in the methods and result sections.

If no control group was investigated, this is a pre-post CI intervention study without a comparison group, and this design has associated biases (placebo effect of the CI) making difficult the outcome interpretation.

De Ridder D et al. Tinnitus and tinnitus disorder: Theoretical and operational definitions (an international multidisciplinary proposal). *Prog Brain Res.* 2021;260:1-25. doi: 10.1016/bs.pbr.2020.12.002. Epub 2021 Feb 1. PMID: 33637213.

Reviewer #4

(Remarks to the Author)

1. The manuscript "A prospective cohort study of cochlear implantation as a treatment for tinnitus in post-lingually deafened adults" addresses the therapeutic impact of cochlear implantation on tinnitus. The authors collected prospective data on 323 hearing-impaired individuals, with or without tinnitus, who were implanted with CI. The results, which align with those of other groups, confirm a significant reduction of loudness as per VAS and Tinnitus Handicap Inventory scores in a proportion of implanted patients. Furthermore, the study determined the positive audiological impact of CI on hearing and the positive influence of CI on patients' anxiety but moderate improvement of tinnitus-induced sleep problems. The prospective design and the relatively long (2 years) follow-up period are commendable.

2. The manuscript is well-written and adds to the existing knowledge about CI's effects on tinnitus. I appreciate the authors' efforts in this regard. Nevertheless, it contains some incorrect claims which must be revised. Moreover, adding some calculations, such as outcome comparison between unilateral and bilateral hearing loss or unilateral and bilateral CI groups, would be recommended.

3. Specific comments:

Abstract:

Line 22: "Cochlear implants, the only safe and effective treatment for deafness..." . this implies that other treatments are not safe and effective. Which treatments are meant? I suggest removing this part of the sentence.

Lines 23-24: "Though anecdotal evidence suggests that cochlear implants suppress tinnitus" – this statement suggests that the proper studies have never been done regarding the subject. This is not the truth, as many clinical studies have been published on that topic in the past 20 years. Later, the authors refer to several published studies, including meta-analysis, which could not be performed when only anecdotal evidence existed. Please revise.

The body text:

I miss a brief definition of tinnitus – please add.

Lines 53 – 54: "However, the relatively small sample size, lack of prospective design, or missing quantitative and systematic outcome measures in these studies have not provided sufficient evidence to make cochlear implantation a labeled indication for the treatment of tinnitus" – there are several claims in this sentence which should justify the study but do not reflect the status of science in the field. First, at least 12 prospective studies were already performed (according to my quick search in PubMed), so it is not the lack of such research but the relatively small number that is the problem. Second, several national and international regulatory agencies are responsible for legalizing indications for any therapy, and such legalization takes time. The sentence (Line 54) sounds like the legalization process (where?) has failed. Was it so?

Lines 91 – 92: The authors write about "tinnitus in both ears" or "tinnitus in one ear." I suggest replacing this with bilateral and unilateral tinnitus. Some patients describe their tinnitus as not being perceived in the ear but in the head; however,

laterality still applies.

It would be appreciated if the authors would add calculations of correlations between tinnitus duration and the implantation outcome, as well as a between-group comparison for unilateral and bilateral tinnitus, unilateral and bilateral hearing impairment, and unilateral and bilateral CI. These additions will further strengthen the manuscript and its potential for impact.

This study included patients between 16 and 18 years of age. According to many guidelines (e.g., [Version 1:](https://nexus.od.nih.gov/all/2018/08/07/human-subjects-and-clinical-trial-glossary-updates/#:~:text=For the purposes of the NIH Policy and,an individual under the age of 18 years), these individuals fall into the category of children. I recommend removing them from the study and recalculating the results.

Reviewers' comments:

Reviewer #1 (Remarks to the Author):

1. Brief summary of the manuscript

The manuscript describes a prospective cohort study assessing the long-term effect of cochlear implantation on tinnitus distress in 323 adult patients with post-lingual deafness. Tinnitus loudness and THI were the primary outcomes. Speech perception, anxiety and sleep quality were the secondary outcomes. These outcomes were measured pre-implantation and at 1, 2, 3, 6, 12, and 24 months after activation. The study showed a significant reduction in tinnitus loudness and distress in the tinnitus cohort. Four patients experienced tinnitus induction after implantation. Speech recognition, anxiety, and sleep difficulties scores significantly decreased at 24 months post-activation compared to baseline. The tinnitus group on average still experience sleep difficulties.

2. Overall impression of the work

The manuscript showed the overall benefit of cochlear implant on tinnitus loudness, distress and related outcomes. The results are not novel (PMCID: PMC9515522, PMCID: PMC10605020), but the study adds value to the field and confirms the findings of the current literature on the effect of cochlear implant on tinnitus. Although the results are clear, the research questions need to be better defined and consistent with the data reported in the manuscript. The manuscript can be improved by providing more details on methods and reporting results (missing data).

Responses: Thank you for your helpful and constructive comments. We have added three references and described their results in the introduction (PMCID: PMC8615734 Assouly et al. 2021; PMC9515522 Assouly et al. 2022; PMC10605020 Deklerck et al. 2023). It is interesting to note that none of these three reports was available at the time of study (2012-2018) and manuscript writing (2020-2021), reinforcing the importance in this emerging topic of research.

Following your suggestions, we have provided detailed methods in the revision. Because we conducted a prospective study with designated research personnel, we were able to collect a complete set of data from 211 tinnitus and 112 non-tinnitus cochlear implant patients. In fact, additional 101 tinnitus and 34 non-tinnitus patients dropped out due to missed follow-up appointments or device failure (see Fig. 1).

3. Specific comments, with recommendations for addressing each comment

Abstract:

1. Line 27: Please specify here the questionnaire / measurement used to assess tinnitus loudness.

Responses: Specified.

2. Line 31: Please specify the questionnaire / measurement used to assess speech, anxiety and sleep.

Responses: **Specified.**

Introduction

1. Line 44: Please define what is tinnitus and add a reference to the definition.

Responses: **Defined.**

2. Line 50: Suggest using the word “electrical” instead of “electric”.

Responses: **Done.**

3. Line 50-57: Only case studies are mentioned here, whereas several works have been published on the topic. Suggest updating this section by referencing to recent systematic reviews on the effect of CI for tinnitus and systematic review on the effect of intracochlear electrical stimulation on tinnitus (e.g. PMC8615734).

Responses: **Updated.**

4. Line 58: Safety is part of the main objective but there is no measurement of safety or reporting of harm and adverse events in the rest of the manuscript. Please remove the word “safety” from the objective or add safety data in the manuscript. Be consistent throughout the manuscript.

Responses: **Changed to “risk” throughout the revision.**

5. Line 65: Same as Line 27. Please specify here the questionnaire / measurement used to assess tinnitus loudness. VAS? Tinnitus matching test?

Responses: **Specified as “loudness estimate on a 0-10 scale”.**

6. Line 67-68: Same as Line 31. Please specify the questionnaire / measurement used to assess speech, anxiety and sleep.

Responses: **Done.**

7. Line 69-70: The secondary aim is unclear. What is the objective? Please rephrase this sentence “The secondary measures would test whether tinnitus affects cochlear implant speech recognition and tinnitus reduction, if any, would alleviate tinnitus-related symptoms.”.

Responses: **Re-phrased as “The secondary endpoints determined whether pre-implant tinnitus and post-implant tinnitus changes affect cochlear-implant speech recognition and tinnitus-related symptoms like anxiety and insomnia”.**

Results

1. Line 74: Was the age of 16 years old chosen as the threshold and not 18 (to select adult)?

Responses: **The age of >16 years old was chosen because children younger than this age would have difficulties in reporting tinnitus. We have added this statement and changed from “adults” to “individuals” in the revision.**

2. Line 84: It could be interested to add information on the cohort hearing profiles (bilateral, unilateral, asymmetrical hearing loss; pure tone average).

Responses: **On average, they were bilateral, symmetrical, sloping severe-to-profound hearing loss (Fig. S1).**

3. Line 91-94: Specify that this is pre-operative data.

Responses: **Specified.**

4. Line 105: Does it mean that patients without tinnitus were asked to fill the VAS and THI as well at each follow-up time point? I would suggest that instead of reporting the average of the NT-cohort, you only report the average score of the patient with induced tinnitus.

Responses: **Yes, they were asked to fill the reports. If we adopted your suggestion, then we would effectively convert the NT-cohort into the T-cohort. It is more appropriate to report the risk of inducing tinnitus in the NT-cohort than the average score of the patients with induced tinnitus. Nevertheless, the small number allowed us to show all individual tinnitus loudness and THI data such that the slightly increased average data (Fig. 2B,C) in the NT-cohort were due to only four individuals, whose data were immediately shown in Fig. 2D,E.**

5. Line 109-116: Was tinnitus induced right after activation for these 4 patients? Or was tinnitus induced at a later follow-up time point?

Responses: **We have added in the revision “Three of them developed tinnitus on the implanted side one month after device activation, while the fourth one developed tinnitus in both the implanted and non-implanted ears immediately after activation.”**

6. Line 117: Please be more precise in reporting results. Instead of “reduced” precise if this reduction is statistically significant and add the difference in score between baseline and 24 months post-implantation.

Responses: **The values are simply descriptive statistics. The differences in scores between the baseline and 24-month post-implantation, and their significance tests, are shown in Fig. 2A and C, and the main text.**

7. Line 132: “Despite the significant difference in baseline between the T- and NT cohorts (50.2 vs. 43.2, Table 1), there was no group difference ($F_{1,321}=2.89$; $P=0.09$).” Please specify that it is post-implantation.

Responses: **Yes, specified.**

8. General comments in the results section:

a. Were there patients reporting total tinnitus suppression?

b. According to Figure 2, there were cases of tinnitus worsening within the T-cohort.

Please report the number of patients reporting worsening.

c. Suggest calculating how many patients reported a total suppression, clinically significant improvement, no significant change and clinically significant worsening. It will give a nice overview of the tinnitus outcomes in the cohort.

Responses: **We have added the following text in the revision: “Specifically, the baseline tinnitus loudness was reduced by 30% or more in 260 out of 350 ears (74.3%) at 24-month post-implantation activation, including, importantly, total suppression or no tinnitus in 127 (36.3%) cases. Furthermore, 29 ears (8.3%) had tinnitus loudness reduction between 0 and 30%. In contrast, 63 ears (17.4%) had increased tinnitus loudness post-implantation, including 55 (15.7%) with 0-30% increase and only 6 (1.7%) with greater than 30% increase in tinnitus loudness.”**

Discussion

1. Line 174: Suggest to better explain the situation CI on / CI off. Please don't use the word "increase" but explain that tinnitus can come back at the same level or higher when not wearing the CI.

Responses: We had measured tinnitus loudness after turning the CI off for 30 minutes. This CI off data are now included as red crosses in Fig. 2B. Indeed, the tinnitus came back from the CI on condition. This CI off condition is now described in the methods, results, and Discussion sections.

2. Line 179: Suggest referencing the systematic review on the effect of intracochlear electrical stimulation on tinnitus (e.g. PMC8615734).

Responses: Referenced as suggested.

3. General comments in the discussion section:

a. Please write a paragraph on the observed changes in tinnitus after cochlear implantation in the study (worsening, induction, reduction) and compare it to the current literature (example of recent literature: PMID: PMC9515522, PMID: PMC10605020). Also, interesting to elaborate on why these changes are observed.

b. Please write a paragraph on the limitations of the study (missing data, VAS loudness not used in CI on and CI off, ...).

c. Several studies and systematic reviews have reported the same observations (e.g. PMID: 26153087), the authors should mention this in their conclusion.

Responses: We have added the following text in the revision: "At an individual level for the T-cohort, 74.3% reported 30% or more reductions in baseline tinnitus loudness, including 36.3% reporting total suppression or no tinnitus, 8.3% showed tinnitus loudness reduction between 0 and 30%, while 17.4% reported increased tinnitus loudness post-implantation."

We think the main limitation of the present study is a lack of a mechanism driven approach and have added the following text in the revision: "While the present study provided strong support for using cochlear implants to treat tinnitus, it did not address the underlying mechanisms."

We have added all three references and the following text in the revision: "While the distributions varied, the present results, especially the relatively high percentage of total suppression and the low incidence of tinnitus induction cases, were consistent with recent studies^{20,21,35}."

Methods

1. Line 190-201: As not everyone is familiar with the hearing loss criteria for CI candidacy in China, I would suggest specifying it. Please specify if it is for bilateral and unilateral hearing loss and the degree of hearing loss for instance. Also interesting to specify why some candidates get one or two CIs, based on which criteria?

Responses: We have added the following text: "All eligible candidates with severe or

more bilateral hearing loss (PTA>80 dB HL)...” At present, only one implant is covered by insurance in China, with the second one being paid for by the patients or their families.

2. Line 190-201: Were there sequential bilateral implantations in the cohort? Did you take this into account?

Responses: The 24 cases were all simultaneous bilateral implantation. We have stated this fact in both Fig. 1 and Fig. 3 legends.

3. Line 190: Remove ‘s’ in the word “effects”.

4. Line 194: Same as Line 74.

Responses: Done.

5. Line 201: Please specify the questionnaire administration type. Was it an online questionnaire or a paper questionnaire? Was it completed by the patient during his/her visit at the clinic or before?

Responses: We added the following text to describe the detailed protocol: “During each visit from pre-surgery to 24-month post-activation, the participants were typically scheduled for a 2-hour appointment at the clinic. The participants first underwent an activation or re-mapping session if needed. With the implant being turned on, they then performed the tinnitus loudness estimation and speech recognition tasks. After which, the implant was turned off for 30 minutes, and while the device was off, they completed the three questionnaires (tinnitus, anxiety and sleep, see sections below) on paper under supervision at the clinic. They were instructed to rest or walk around if they finished the questionnaires shorter than the 30 minutes. After this 30-minute device off period, they performed the tinnitus loudness estimation task while the device was still off.”

6. Line 206: If the tinnitus loudness estimate is a Visual Analogue Scale (VAS), please mention it and keep this wording throughout the manuscript for consistency.

Responses: We didn’t use a visual analog scale, instead a numerical scale from 0 to 10 with a verbal description stating that “0 representing inaudible and 10 being an unbearably loud sound”.

7. Was the VAS loudness used with the CI was on or off or in both conditions? Please specify.

Responses: We have done both, including measuring tinnitus loudness after turning off the CI for 30 minutes during each visit. The data are presented as crosses in Fig. 2B and described in the revision.

8. Line 215: Suggest using the standard Minimal Clinically Important Difference defined in the literature as specified in the publications: 7 points for THI (PMID: 21493265) and 1.5 for VAS loudness (PMID: 22846637).

Responses: Used as suggested, with both publications being referenced in the revision.

9. Line 223: Please specify with which software/tool you perform the statistical analysis.

Responses: GraphPad Prism version 10.0 for Windows, which is now described in the revision.

10. Line 224-228: Did you test the normal distribution of your data?

Responses: We didn’t check all of them. We have now found that many in Table 1 did not pass the normality test. Therefore, non-parametric Mann-Whitney U tests instead of t-tests are used in the revision. Table 1 is now revised with the new tests. but importantly, none of the original results is changed.

11. Line 224-228: How did you manage missing data? Please report missing data at every follow-up time point.

Responses: We lost 89 cases (66 in the T-cohort and 23 in the NT-cohort) due to missed appointments. The remaining cases did not have any missing data.

Reviewer #2 (Remarks to the Author):

The authors provide a study that tracks and assesses quite a large number of cochlear implant patients over time in terms of their tinnitus, as well as other outcomes of speech perception, sleep and anxiety. The importance and high impact of the study is in the large numbers possible. Overall, the results are encouraging and important in showing that cochlear implant stimulation over time can improve tinnitus symptoms, as well as other outcomes that are expected based on previous studies. The paper is generally written in a clear way, however some key details are missing. Other suggestions to improve the manuscript are provided below.

Responses: **Thank you for your helpful and constructive comments.**

>>There are multiple places where sentences and wording can be improved; please review carefully to catch all of them. Some examples are listed below:

-Abstract: Though anecdotal evidence suggests that cochlear implants suppress tinnitus, this utility is not yet established. In a decade-long study, we examined longitudinal effects of cochlear implants on tinnitus in 323 post-lingually deafened adults including 211 with pre-existing tinnitus and 112 without [tinnitus].

Responses: **Done.**

-Abstract: Prior to implantation, the tinnitus cohort had lower speech recognition, higher anxiety and poorer sleep quality than the non-tinnitus cohort, [in which] the 24-month implant usage eliminated this group difference in speech and anxiety measures.

Responses: **This sentence has been broken into two sentences in the revision.**

-Introduction: Last 3 sentences say “would” but already happened so like previous sentences use past tense.

Responses: **Done.**

-Results, update to:

From January 1, 2012 to December 31, 2018, a total of 2998 patients received cochlear implants at the Chinese PLA General Hospital. 2141 were excluded based on either the age (younger than 16 years) or pre-lingual deafness criterion; the remaining 577 were then screened for tinnitus (Fig. 1). The screening showed that 431 of them (74.7%) had tinnitus, in which 119 individuals did not meet the inclusion criterion and 101 dropped out during the study. Thus, the tinnitus group (T-cohort) had a sample size of 211, with 187 of them receiving one cochlear implant while 24 receiving bilateral implants. The screening also showed that 146 (25.3%) had no tinnitus, in which 34 of them dropped out during the study. The no-tinnitus group (NT-cohort) had a sample size of 112, with 105 of them receiving one and 7 receiving two cochlear implants.

Responses: **Updated as suggested. Thank you for your careful proofread.**

-Results, update to:

There were no significant group differences in gender, duration, etiology of deafness, cochlear implant brand and ears implanted (Table 1). There were also no significant group differences in the pre-surgical audiograms, showing similarly symmetrical, sloping severe-to-profound hearing loss (Fig. S1).

Responses: **Updated as suggested.**

-Results section; formatting of CI can be improved, such as:

(total reduction= -21.2; 95% CI of, -24.5 to 0-17.9; $P < 0.0001$...) where need some spacings and filler words; also confusing to include % for mean value but not CI so remove or provide completely.

Responses: **% for mean value has been removed as suggested. Other changes followed the format suggested by Communication Medicine.**

-Discussion, 3rd paragraph: Using the word “super-large effect size” is a bit odd; consider using “quite large effect size” or something more formal.

Responses: **Done.**

-Discussion last sentence could be improved to:

A solution for this group of tinnitus patients is to provide stepwise intervention from sound therapy to non-invasive electric stimulation, after which cochlear implantation using an atraumatic electrode array and a soft surgical approach to preserve hearing could be possible as a last resort. The present findings pave the way for cochlear implantation to be an alternative and effective treatment for not only deafness but also tinnitus.

Responses: **Done.**

-Figure 1 title could be more reflective of content such as, “participant numbers and groups across study”, since not really showing study design (actual design and timeline is shown in Figure 1A).

Responses: **Done.**

>>One major improvement needed is in more rigorously describing the selection of the statistical analyses performed, such as why Chi-squared tests and in justifying normality assumption for the t-tests. Also, it wasn't sufficiently described how multiple comparisons issue was addressed, such as which endpoints were pre-specified or in what order, or that the multiple comparisons were addressed. Please provide further details to clarify and justify statistical approach. For example, claiming a significant group difference in baseline secondary outcome measures of T-cohort having lower speech recognition than NT-cohort with a $p = 0.02$ is not convincing when so many comparisons are being done.

Responses: **We have added details in the Methods section to more clearly describe our**

statistical analyses. One reason for the confusion was a lack of clear definition between baseline and endpoint measures. We have now clarified which variables are "endpoints" (i.e., longitudinal primary and secondary measures). Other variables compared are classified as outcome variables rather than endpoints (i.e., demographic and baseline comparisons). We added information to describe that data with continuous variables were first tested for normality using a Shapiro-Wilk test, and then tested with t-tests if data were normally distributed, otherwise a non-parametric Mann-Whitney U test was used. We also explained that Chi-squared tests were used when comparing categorical variables. We clarified that adjusted p-values were reported using a Dunnett's test to correct for multiple comparisons across longitudinal data.

>>Another major improvement needed is in better describing the study design and procedures in the Methods section, such as how were patients identified, what was done at each visit and what order of assessments, and even how were tinnitus characteristics determined (e.g., unilateral vs. bilateral, tinnitus type, etc.). Were they just asked questions or written on paper or a tablet, etc. Please provide more details in the first part of the Methods section for the reader to follow in detail what was done during the study. Also include the full inclusion/exclusion list in the Methods.

Responses: We have added the following text in the Methods section to describe the detailed protocol: "During each visit from pre-surgery to 24-month post-activation, the participants were typically scheduled for a 2-hour appointment at the clinic. The participants first underwent an activation or re-mapping session if needed. With the implant being turned on, they then performed the tinnitus loudness estimation and speech recognition tasks. After which, the implant was turned off for 30 minutes, and while the device was off, they completed the three questionnaires (tinnitus, anxiety and sleep, see sections below) on paper under supervision at the clinic. They were instructed to rest or walk around if they finished the questionnaires shorter than the 30 minutes. After this 30-minute device off period, they performed the tinnitus loudness estimation task while the device was still off."

We have also added a line in the Table 2 title to explicitly state that "All tinnitus characteristics were based on self-reports by the participants who filled a tinnitus survey form on paper."

>>Please provide in the Methods the ethics approval information and number and that study was performed in accordance with those requirements.

Responses: Done.

>>The results are quite positive and convincing that cochlear implant improves tinnitus over time, and the results are well presented in the figures and greatly appreciate that the authors provided the individual participant data for tinnitus and loudness in Figure 2, which is convincing.

Responses: Thanks.

>>Note that MCID for THI is 7 points and not 6-7 points. Please update in Methods.
Responses: **Updated.**

>>Multiple statements in Discussion need to be revised since they are not fully valid or clear.

-In paragraph 2, authors make strong claims that cochlear implant was “greater and faster” than other approaches. A major concern is the authors did not accurately or sufficiently cite data from previous studies. The vagus nerve study reported 17.7 for THI in terms of percentages and not points so those values need to be updated if comparing in points. The bimodal sound and tongue stimulation is based on treatment data for 12 weeks and not 6 months, in which there is a more recent study showing ~20 points by 12 weeks (actual values can be obtained from Conlon, B., et al. Different bimodal neuromodulation settings reduce tinnitus symptoms in a large randomized trial. *Sci Rep* 12, 10845 (2022). So those data need to be updated. Also, there are several sound therapy studies showing comparable or even better results at similar rates (see papers by James Henry). Even if they don’t use THI and they use TFI or other outcome measures, it doesn’t mean they can be ignored for comparison (e.g., see Henry, J. A., et al., Tinnitus management: Randomized controlled trial comparing extended-wear hearing aids, conventional hearing aids, and combination instruments. *Journal of the American Academy of Audiology*, 28(06), 546-561). Based on the available literature, it appears cochlear implant can provide large improvements in tinnitus symptoms that can be better or comparable to other leading treatments that is as fast or in some cases faster for other types. The comparisons are more nuanced than the broad strong claim the authors make; please more carefully compare and explain the outcomes relative to many more studies to be valid.

Responses: **We have revised this paragraph as suggested by softening the claim, more accurately describing the results, and adding two new references.**

-Last paragraph the authors claim “little research has been done” on cochlear implant effects on tinnitus improvements. This is not fully true in that multiple studies on this topic especially in unilateral deafness/tinnitus patients. Please adjust wording and also cite some of these previous studies to put authors results into more comprehensive context.

Responses: **The claim “little research has been done” has been removed and references have been added in the revision (see also responses to reviewer 1’s comments).**

-The authors should further think about the intention of the cochlear implant approach in their statement: “The present findings pave the way for cochlear implantation to be a safe and effective treatment for not only deafness but also tinnitus.” Are the authors claiming individuals get implanted with a cochlear implant solely for tinnitus, even in individuals with some hearing remaining in that ear side?” That seems then there are major risks that can be possible such as losing their residual hearing and needs to be further discussed pros/cons. If only for those with sufficient hearing loss, it would be

helpful to put that conclusion in the context the types of patients assessed in this study and how much hearing loss they had to identify generalizability of results to those with less hearing loss.

Responses: Thank you for raising these important questions. The last sentence in Discussion has been revised to: "If future implantation causes no or minimal hearing loss, then the cochlear implant can be an effective treatment for not only deafness but also tinnitus, even for those tinnitus sufferers who have significant residual or normal hearing."

Reviewer #3 (Remarks to the Author):

This is a clinical trial on the effect of cochlear implants on a 211 hearing loss and tinnitus cohort. I have some concerns with the study design since the inclusion criteria are not clearly defined and there is no control group.

The experimental group included patients with > 3months with chronic tinnitus; however, there is no information about the psychoacoustic (tinnitus frequency, loudness, MML, residual inhibition), limited information about psychometric (THI score range 2-98) and most important hearing levels for low and high frequencies. It is expected that many of these variables can explain the outcome of the trial.

The inclusion criteria should be precisely defined: tinnitus type, defined THI range and defined threshold hearing. The data are presented as any tinnitus type and the authors should differentiate any tinnitus from tinnitus disorder (see reference), since they are clinically different in term of management and interventions.

According to the information provided, no control group (tinnitus patients without CI) or randomization method was used. Please, try to clarify this in the methods and result sections. If no control group was investigated, this is a pre-post CI intervention study without a comparison group, and this design has associated biases (placebo effect of the CI) making difficult the outcome interpretation.

De Ridder D et al. Tinnitus and tinnitus disorder: Theoretical and operational definitions (an international multidisciplinary proposal). Prog Brain Res. 2021;260:1-25. doi: 10.1016/bs.pbr.2020.12.002. Epub 2021 Feb 1. PMID: 33637213.

Responses: Thank you for your helpful and constructive comments. The present study is not a clinical trial but a prospective cohort study, thus lacking control and randomization.

Because all tinnitus patients in the present study had bilateral severe-to-profound hearing, it would be difficult, if not impossible, to collect the stated psychoacoustic measures.

We have included the complete inclusion criteria in the revision. Individual tinnitus loudness and THI data can be seen in Fig. 2D and E.

We agree on the importance in differentiating between tinnitus and tinnitus disorder and have added the following text (and the reference) in the revised Methods section: "In contrast, the global THI measure cannot follow the short-term or ear-specific changes in tinnitus, but rather reflect the functional, emotional and catastrophic aspect of tinnitus disorder⁴⁹."

Reviewer #4 (Remarks to the Author):

1. The manuscript “A prospective cohort study of cochlear implantation as a treatment for tinnitus in post-lingually deafened adults” addresses the therapeutic impact of cochlear implantation on tinnitus. The authors collected prospective data on 323 hearing-impaired individuals, with or without tinnitus, who were implanted with CI. The results, which align with those of other groups, confirm a significant reduction of loudness as per VAS and Tinnitus Handicap Inventory scores in a proportion of implanted patients. Furthermore, the study determined the positive audiological impact of CI on hearing and the positive influence of CI on patients' anxiety but moderate improvement of tinnitus-induced sleep problems. The prospective design and the relatively long (2 years) follow-up period are commendable.

Responses: Thank you for your helpful and constructive comments.

2. The manuscript is well-written and adds to the existing knowledge about CI's effects on tinnitus. I appreciate the authors' efforts in this regard. Nevertheless, it contains some incorrect claims which must be revised. Moreover, adding some calculations, such as outcome comparison between unilateral and bilateral hearing loss or unilateral and bilateral CI groups, would be recommended.

Responses: We have addressed incorrect claims identified by you and other reviewers (see specific responses). Thanks to your recommendation, we have added a section on the interactions between unilateral/bilateral CIs and tinnitus (see also Fig. 3 and Table S3 in the revision). The data showed a clear pattern that cochlear implantation produces more suppression when the implant and tinnitus are on the same side than the opposite side.

3. Specific comments:

Abstract:

Line 22: “Cochlear implants, the only safe and effective treatment for deafness...” . this implies that other treatments are not safe and effective. Which treatments are meant? I suggest removing this part of the sentence.

Responses: Removed as suggested.

Lines 23-24: “Though anecdotal evidence suggests that cochlear implants suppress tinnitus” – this statement suggests that the proper studies have never been done regarding the subject. This is not the truth, as many clinical studies have been published on that topic in the past 20 years. Later, the authors refer to several published studies, including meta-analysis, which could not be performed when only anecdotal evidence existed. Please revise.

Responses: Revised as suggested.

The body text:

I miss a brief definition of tinnitus – please add.

Responses: **Added.**

Lines 53 – 54: “However, the relatively small sample size, lack of prospective design, or missing quantitative and systematic outcome measures in these studies have not provided sufficient evidence to make cochlear implantation a labeled indication for the treatment of tinnitus” – there are several claims in this sentence which should justify the study but do not reflect the status of science in the field. First, at least 12 prospective studies were already performed (according to my quick search in PubMed), so it is not the lack of such research but the relatively small number that is the problem. Second, several national and international regulatory agencies are responsible for legalizing indications for any therapy, and such legalization takes time. The sentence (Line 54) sounds like the legalization process (where?) has failed. Was it so?

Responses: **Revised and several references have been added (see also responses to reviewers 1 and 2’s comments). The sentence simply stated a fact.**

Lines 91 – 92: The authors write about “tinnitus in both ears” or “tinnitus in one ear.” I suggest replacing this with bilateral and unilateral tinnitus. Some patients describe their tinnitus as not being perceived in the ear but in the head; however, laterality still applies.

Responses: **Done.**

It would be appreciated if the authors would add calculations of correlations between tinnitus duration and the implantation outcome, as well as a between-group comparison for unilateral and bilateral tinnitus, unilateral and bilateral hearing impairment, and unilateral and bilateral CI. These additions will further strengthen the manuscript and its potential for impact.

Responses: **We did correlations between tinnitus duration and implantation outcomes (see below) and found nothing significant.**

Pearson's r-value, p-value, respectively, for THI:

Month 1: -0.01, 0.9

Month 2: -0.01, 0.94

Month 3: -0.06, 0.45

Month 6: -0.07, 0.35

Month 12: -0.04, 0.60

Month 24: -0.05, 0.45

Pearson's r-value, p-value, respectively, for tinnitus loudness:

Month 1: 0.04, 0.62

Month 2: 0.05, 0.51

Month 3: 0.06, 0.44
Month 6: 0.06, 0.40
Month 12: 0.08, 0.30
Month 24: 0.06, 0.44

We did add the interactions between the device and tinnitus laterality as suggested and found a significant lateralization effect as a result (see Fig. 3 in the revision). Thank you.

This study included patients between 16 and 18 years of age. According to many guidelines (e.g., [Responses: We have removed the word “adults” and changed to “individuals” in the revision.](https://nexus.od.nih.gov/all/2018/08/07/human-subjects-and-clinical-trial-glossary-updates/#:~:text=For the purposes of the NIH Policy and,an individual under the age of 18 years), these individuals fall into the category of children. I recommend removing them from the study and recalculating the results.<div data-bbox=)

Reviewers' comments:

Reviewer #1 (Remarks to the Author):

1. Overall impression of the work

The manuscript has been improved and all my comments have been addressed. Please find below some minor additional comments on the current version of the manuscript.

Responses: Thank you again for your careful and helpful comments.

2. Specific comments, with recommendations for addressing each comment

Introduction

“However, the relatively small sample size, lack of prospective design, or missing quantitative and systematic outcome measures in these studies have not provided sufficient evidence to make cochlear implantation a labeled indication for the treatment of tinnitus”.

Please correct this sentence as there are studies with a prospective design and with systematic outcome measures, that you just mentioned before in the text. You could highlight the lack of a ‘control’ group, i.e. group without tinnitus. This will highlight the novelty of your study compared to others.

Responses: We have deleted “the relatively small sample size, lack of prospective design, or missing quantitative and systematic outcome measures in” as suggested.

Results

“Specifically, the baseline tinnitus loudness was reduced by 30% or more in 260 out of 350 ears (74.3%) at 24-month post-implantation activation, including, importantly, total suppression or no tinnitus in 127 (36.3%) cases. Furthermore, 29 ears (8.3%) had tinnitus loudness reduction between 0 and 30%. In contrast, 63 ears (17.4%) had increased tinnitus loudness post-implantation, including 55 (15.7%) with 0-30% increase and only 6 (1.7%) with greater than 30% increase in tinnitus loudness.”

Suggest describing the results for individuals (if possible) rather than for ears or adding similar information (percentage by category) for individuals. In this way, the study will be aligned with other studies reporting changes by individuals and will be compared with all the studies and used for future reviews/meta-analysis.

Discussion

“At an individual level for the Tcohort, 74.3% reported 30% or more reductions in baseline tinnitus loudness, including 36.3% reporting total suppression or no tinnitus, 8.3% showed tinnitus loudness reduction between 0 and 30%, while 17.4% reported increased tinnitus loudness post-implantation.”

In the discussion, you mention the same percentage as that mentioned in the results section regarding the percentage per ear, but you now say that it is at individual level. Please correct these percentages so that they are representative of the individual level (if possible).

Responses: Because the only reasonable individual measure is THI, we now report the THI related data at the individual level in both the Results section and Discussion:

The Results section: “At 24-month post-implantation activation, the baseline THI was reduced by 31% or more in 130 out of 211 participants (61.6%) and by 0-30% in 60 participants (28.4%); the THI was increased in only 21 participants (10.0%), including 19 (9.0%) with 0-30% increase and 2 (0.9%) with greater than 30% increase.”

The Discussion section: “At an individual level for the T-cohort, 90.0% reported decreased THI scores, including 61.6% reporting greater than 30% reduction and 28.4% reporting 0-30%, while only 10.0% reported increased THI post-implantation.”

Figure 3

Please correct the word ‘tinniut’ by ‘tinnitus’ in the y-axis of Figure 3.

Responses: Done.

Reviewer #2 (Remarks to the Author):

The authors have addressed most of my concerns.

Responses: Thank you for your helpful and constructive comments.

Further details on the statistics/data analysis part have been provided including clarifying the primary and secondary endpoints; please also clarify in the abstract which are the primary and which are the secondary endpoints as was done in a helpful way in the Introduction.

Responses: We have added the following sentences in the revised abstract: “The primary endpoints were tinnitus loudness and tinnitus handicap inventory. The secondary endpoints were speech recognition, anxiety and sleep quality.”

The authors still have mixed up and incorrectly cited tinnitus change scores/values and types in the Discussion 2nd paragraph. Authors must carefully review those cited papers and the results they mention since some are still incorrect; please reread the my previous comments in how to fix some of them to be valid since several were missed and not addressed as requested previously.

Responses: We apologized for mixing up the bimodal stimulation study with the vagus nerve study. We have corrected this and other mistakes and rewritten this sentence in the revised Discussion:

“Emerging treatments such as paired tone and vagus nerve stimulation reduced tinnitus loudness by -0.6 and THI by -20.2 over 3 months³⁹, while bimodal sound and tongue stimulation reduced THI by -13.5 to -14.6 over a 12-week treatment period⁴⁰, which was further reduced to -21.2 by changing s parameter settings in the second 6-week period⁴¹.”